# DiffuReason: Enhancing Reasoning Ability for Diffusion Language Models via Monte Carlo Tree Search

Yiping Song [* 1]   Jinyu You [* 1]   Zhiliang Tian [1]   Jinsong Su [2]   Minlie Huang [3]   Chenping Hou [1]

## Abstract

Auto-Regressive (AR) models with Monte Carlo Tree Search (MCTS) are a dominant paradigm for achieving "System 2" reasoning. However, this approach suffers from significant latency due to the serial, token-by-token generation mechanism of AR models. In contrast, Diffusion Large Language Models (dLLMs) offer inherent speed advantages via parallel sequence generation, yet they often struggle with accuracy in complex reasoning due to a lack of rigorous search, evaluation, and revision capabilities. Directly applying MCTS to diffusion models faces architectural barriers, since the denoising generation process lacks the discrete decision steps that naturally accommodate tree search. To retain efficiency while improving the reasoning ability, we propose DiffuReason, a Monte Carlo tree search reasoning algorithm for diffusion models. By modeling the generation process as a Markov Decision Process (MDP), DiffuReason discretizes the continuous diffusion flow into searchable thought blocks. During the reverse generation process, DiffuReason recursively performs four MCTS-style stages: select the best node (block), expand to obtain candidate nodes, simulate to evaluate node values, and revise the unsatisfactory nodes. Experiments on mathematical reasoning benchmarks demonstrate that DiffuReason significantly improves the reasoning ability of diffusion models, and achieves superior balance of accuracy and efficiency even compared with auto-regressive models.

*Equal contribution   [1]National University of Defense Technology, Changsha, China [2]Xiamen University, Xiamen, China [3]Tsinghua University, Beijing, China. Correspondence to: Minlie Huang <aihuang@tsinghua.edu.cn>, Chenping Hou <hcp-nudt@hotmail.com>.

*Proceedings of the 43rd International Conference on Machine Learning*, Seoul, South Korea. PMLR 306, 2026. Copyright 2026 by the author(s).

## 1. Introduction

With the launch of the OpenAI o1 (Wu et al., 2024), o3 (El-Kishky et al., 2025), and DeepSeek R1 (Guo et al., 2025), the academic community observes that "slow thinking" mechanisms—in which models internally perform multi-step chain-of-thought reasoning—show significant potential in solving complex cognitive tasks such as mathematical problems and programming. Although reasoning accuracy remains a core metric of model intelligence and a major focus for researchers, reasoning efficiency (such as response latency and computational resource consumption) has also become a crucial factor determining the practical value of models, directly impacting user acceptance and depth of usage. Therefore, optimizing reasoning efficiency while maintaining high-level reasoning performance is a pressing and important research topic in the current field of generative artificial intelligence (Cao, 2023).

Auto-Regressive (AR) models are the dominant paradigm in language generation, characterized by their causal, token-by-token generation mechanism. To enable deeper logical reasoning within this architecture, researchers integrate "slow thinking" algorithms such as Monte Carlo Tree Search (MCTS) (Yao et al., 2023; Chaslot et al., 2008; Pan et al., 2025; Gan et al., 2025). Monte Carlo Tree Search (MCTS) comprises four core steps (Selection, Expansion, Simulation, and Backpropagation) to achieve three critical functionalities: Searching: constructing a tree-structured exploration space that supports diversified search; Simulation: performing dynamic value estimation for each in-process node in the tree; Revision: iteratively refining node value estimates via simulations to guide future exploration. However, due to the inherent serial generation nature of AR models, MCTS suffers from a low efficiency problem: complex problems require a large number of simulations, and each simulation generates words token-by-token; propagating node values back requires additional computation.

In contrast, Diffusion Large Language Models (Nie et al., 2025a; Liu et al., 2025) (dLLMs), particularly mask Diffusion Models (MDMs) (Shi et al., 2024; Zheng et al., 2024), break causal dependencies, allowing parallel sampling of multiple tokens throughout the sequence within each diffusion step (Austin et al., 2021; Nie et al., 2025b). The-

oretically, this parallel generation offers the potential for rapid inference. Although in practice, "diffusion models are faster" holds only when measuring local approximation quality; the advantage may vanish when pursuing global precision (Feng et al., 2025). In reasoning tasks, which require the demand for both global planning and local precision, the efficiency advantage of diffusion models cannot be realized. Thus, a key challenge for diffusion models lies in diffusion model is how to take advantage of their logical planning capabilities of diffusion models while maintaining their parallel advantages, thereby achieving high accuracy similar to MCTS-enhanced AR models.

To enhance the reasoning capability of diffusion models, existing diffusion-based models attempt to mimic the Chain-of-Thought (CoT) for "slow thinking". Some approaches (Ye et al., 2024) use a given chain-of-thought as a supervisory signal to constrain the output at each denoising step; others bypass CoT as supervision and instead design more self-correction mechanisms, such as resampling within a sliding window (Cao et al., 2026) or re-masking low-confidence regions (Berrayana et al., 2025). These methods align the time steps of the reverse denoising process with each step of CoT, enabling the diffusion model to think step-by-step and allowing for local corrections within the reasoning path. However, such local error correction can only fix mistakes within the existing line of thought and cannot explore multiple diverse reasoning alternatives, which remains unsatisfactory compared to MCTS-enhanced AR models.

Directly integrating MCTS with diffusion models appears to be a natural solution. However, transposing the classic MCTS workflow to diffusion models faces fundamental architectural barriers: MCTS relies on discrete, sequential state transitions, whereas diffusion generation is intrinsically a continuous, full-sequence progressive denoising process. This misalignment between discrete planning and continuous generation renders traditional tree search algorithms unable to directly interface with the diffusion flow. ·To address the above challenges, this paper proposes DiffuReason[1], which effectively integrates the accuracy of slow thinking from auto-regressive models with the efficient fast generation of diffusion models. DiffuReason fully inherits the three core capabilities of MCTS: Searching: it discretizes the continuous diffusion stream into semantically complete "thought blocks", constructing a discrete state space for tree search; Simulation: it performs Lookahead Rollout and fast generation mechanism to quickly estimate the long-term value of reasoning paths; Revision: it introduces an In-place Revision strategy and allows re-sampling trigger for unsatisfied reasoning node. Through the MCTS-style four steps (Selection, Expansion, Simulation, and Revision), DiffuReason achieves fine-grained control over the diffusion reasoning process and maintains a superior balance between accuracy and efficiency. The main contributions of are:

- We propose DiffuReason, an MCTS-style reasoning framework for diffusion models. It combines the strengths of the high reasoning accuracy of auto-regressive models and the high efficiency of diffusion models.

- Catering to the denoising generation scheme of the diffusion model, we propose four reasoning steps (Selection, Expansion, Simulation, and Revision) that can effectively inherit the searching, simulation, and revision capabilities of MCTS to achieve precise reasoning.

- Experiments demonstrate that DiffuReason outperforms diffusion models in reasoning quality while being faster than autoregressive models.

## 2. Related Work

### 2.1. Monte Carlo Tree Search.

To transcend the limitations of "System 1" intuitive reasoning, researchers endow Auto-Regressive (AR) models with "System 2" capabilities by integrating planning algorithms such as Monte Carlo Tree Search (MCTS) (Chaslot et al., 2008; Pan et al., 2025; Gan et al., 2025). MCTS shows superior performance for complex reasoning due to its strong search, evaluation, and revision ability. MCTS-based methods decompose reasoning into discrete thought units and use BFS or DFS for structural exploration, such as Chain-of-Thought (CoT) (Wei et al., 2022a) and Tree of Thoughts (ToT) (Yao et al., 2023). Researchers further use LLM to evaluate discrete thoughts, then revise the reasoning process. For example, RAP (Hao et al., 2023) treats LLMs as both a World Model and an agent, employing MCTS to perform lookahead simulations within a vast reasoning space; AlphaMath (Chen et al., 2024) leverages MCTS to generate high-quality process supervision data, training a Value Model to guide the search for reasoning paths; other works enhance MCTS thinking diversity through self-reflection (Zhang et al., 2024; 2025). Despite the superior accuracy, MCTS-based methods suffer from lower efficiency for two reasons: first, existing MCTS-based methods are based on auto-regressive models, and the sequential word-by-word generation inherent in auto-regressive methods is naturally slower; second, in the simulation process of MCTS, the final result can only be obtained through step-by-step generation, and the complex backtracking mechanism further increases time consumption.

---

[1]Our code is available at: https://github.com/shanglicat1/Diffusion_reason

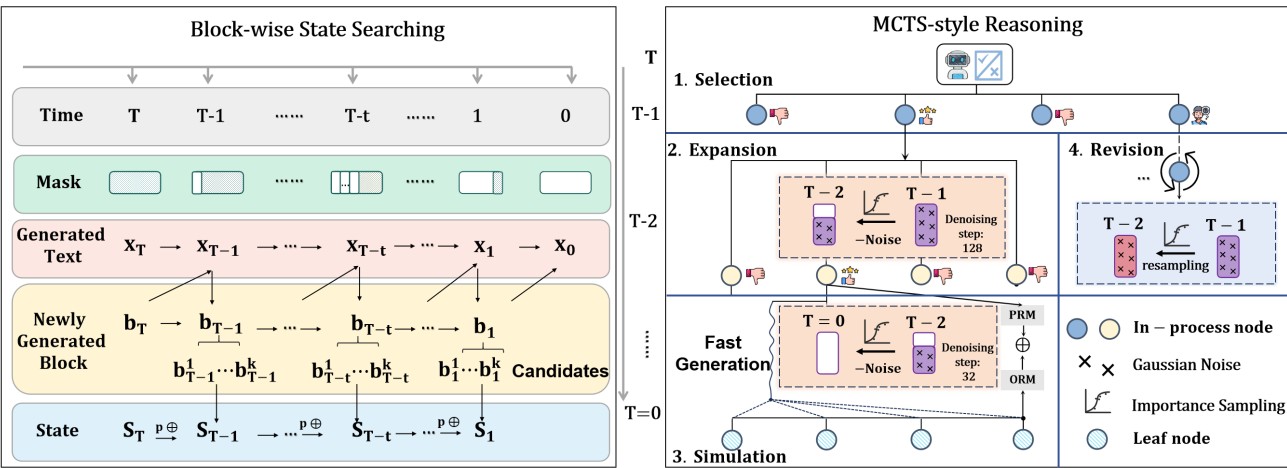

*Figure 1.* DiffuReason: The left panel presents how to formulate the diffusion generation process as a block-wise searching scheme, and the right panel presents the four steps of searching, namely Selection, Expansion, Simulation, and Revision.

## 2.2. Reasoning in Diffusion Models.

With the application of diffusion models in text generation, eliciting their latent reasoning capabilities has become a key direction. One mainstream methods employ Post-training strategies to improve the reasoning ability. D1 (Zhao et al., 2025) represents this direction, mirroring OpenAI's o1 approach by proposing Reinforcement Learning (RL) combined with Supervised Fine-Tuning (SFT) to expand the reasoning capabilities of diffusion LLMs. Another mainstream methods abandon post training and focuses on aligning the Chain-of-Thought (CoT) with the step-by-step denoising process of diffusion models, enabling iterative thinking and correction. For example, Diffusion of Thought (DoT) (Ye et al., 2024) treats each thinking step of CoT as each time step of diffusion reverse generation, and allows for self-correction by adjusting all words at each step of the denoising process. DiffCoT (Cao et al., 2026) sets a sliding window to perform small-scale corrections on the CoT thinking chain. Although diffusion-style thinking is introduced at the step level, DiffCoT still maintains an auto-regressive generation property at the token level. These methods lack node evaluation to guide the precise correction of CoT, so R3 (Berrayana et al., 2025) proposes to use a Process Reward Model (PRM) to evaluate generated text and perform remasking and regeneration on low-confidence regions based on scores. However, due to the inherently linear nature of CoT itself, existing methods are unable to motivate diverse thought processes using a tree structure like Monte Carlo Tree Search (MCTS) does. In contrast, DiffuReason relies solely on Test-time scaling (Snell et al., 2024) of pre-trained diffusion models, and is able to perform tree search, deep simulation, and precise revision, which, in our opinion, is the essence of the superiority of MCTS.

## 3. Method

In this section, we detail DiffuReason (shown in Figure 1), an inference-time planning framework designed specifically for diffusion large language models. DiffuReason formalizes the reasoning process as a Block-wise State Search and formulates a new MCTS framework incorporating Selection, Expansion, Simulation, and Revision.

### 3.1. Reasoning as Block-wise State Searching

We formalize the complex mathematical reasoning process as a Markov Decision Process (MDP) over a discrete state space. We first introduce the basic generation model, then define the searching space of MDP, and finally illustrate the searching formulation.

#### 3.1.1. MASKED GENERATIVE DIFFUSION.

Unlike Auto-Regressive (AR) models that generate token-by-token based on unidirectional causal masks, we employ Masked Diffusion Models (MDMs, such as LLaDA (Nie et al., 2025b)) as the generator $G_\theta$. MDMs learn a bidirectional denoising process aimed at recovering the original discrete sequence $x_0$ from a fully masked sequence.

In the forward process, for a discrete sequence $x_0$, each token $x_0^i$ is independently replaced by a mask token [M] with a probability $t \in [0, 1]$. The transition probability is formulated as:

$$q(x_t^i|x_0^i) = (1 - t) \cdot \mathbb{I}[x_t^i = x_0^i] + t \cdot \mathbb{I}[x_t^i = [M]] \quad (1)$$

where $\mathbb{I}[\cdot]$ is the indicator function.

At the reverse generation (inference) time, starting from a fully masked sequence $x_{t=1} = [[M], \ldots, [M]]$, the diffu-

sion model $G_\theta$ predicts the probability of unmasked tokens $x_0$ given the current masked sequence $x_t$. To achieve block-wise generation, a deterministic mask $M_t$ is used to control which tokens are revealed at time $t$. For the tokens to be generated, we sample from the categorical distribution using the Gumbel-max trick:

$$\hat{x}_0^i = \arg\max_{v \in \mathcal{V}} \left(\log p_\theta(v|x_t) + \tau \cdot g_v\right) \tag{2}$$

where $\mathcal{V}$ is the vocabulary, $g_v \sim \text{Gumbel}(0,1)$ is the standard Gumbel noise, and $\tau$ is the temperature controlling the stochasticity of the generation.

### 3.1.2. REASONING STATE SPACE.

To address planning issues in long-chain reasoning, we model the generation process as a search task in a state space $\mathcal{G} = (\mathcal{S}, \mathcal{A}, \mathcal{T}, \mathcal{R})$ (reverse generation from timestep $T$ to timestep 0):

- State Space $\mathcal{S}$: State $s_{T-t} = [p, b_T, \ldots, b_{T-t}]$ represents the currently generated reasoning path, where $p$ is the problem prompt and $b_{T-t}$ are the $T - t$-th time generated thought blocks.

- Action Space $\mathcal{A}$: To adapt to MCTS and compress search depth, we define action $a_{T-t}$ as generating the next complete Thought Block $b_{T-(t+1)}$, rather than a single token. The action space is implicitly defined by the diffusion generator.

- Transition (Transition) $\mathcal{T}$: State transition $s_{T-(t+1)} = s_{T-t} \cup b_{T-(t+1))}$ is executed by the generator $G_\theta(s_t)$.

- Value (Value) $\mathcal{R}$: We introduce a Process Reward Model (PRM) as the value function $V(s) \in [0, 1]$ to evaluate the logical rigor of the current reasoning path.

### 3.1.3. SEARCHING BLOCK FORMULATION

Let the target prediction $x_0$ be a vector of length $L$. It is generated from pure noise $x_T$ over $T$ time steps, with the time steps proceeding from $T$ to 0 (following the reverse generation principle of diffusion models). The entire length $L$ is divided into $T$ blocks, each of length $\frac{L}{T}$. One block is generated at each time step, with each block being generated based on the previous blocks. To ensure that the content generated at each time step adheres to the above setup, we use a mask $M$ to control this process as illustrated in Section 3.1.1. Specifically, at each time step, the first $L - \frac{L}{T} \cdot t$ dimensions of the mask vector are set to 1, and the remaining $\frac{L}{T} \cdot t$ dimensions are set to 0:

$$M_t = [\underbrace{1, \cdots, 1}_{L-\frac{L}{T} \cdot t}; \underbrace{0, \cdots, 0}_{\frac{L}{T} \cdot t}] \tag{3}$$

$$[b_T, b_{T-1} \cdots, b_{T-t}] = \hat{x}_0 \odot M_t \tag{4}$$

where $\odot$ denotes the discrete masking operation, which preserves the predicted tokens $\hat{x}_0^i$ where $M_t^i = 1$, and forces the tokens to remain [M] where $M_t^i = 0$. By regarding each state $s_{T-t} = [p, b_T, \ldots, b_{T-t}]$ as a node, DiffuReason efficiently predicts multiple semantically coherent text blocks in parallel (Block-wise Generation), and it is able to perform structural searching as a tree in diffusion models, the same as MCTS in auto-regressive models.

### 3.2. MCTS-style Reasoning

Similar to MCTS, DiffuReason also has four critical stages: Selection for choosing a child node, Expansion for generating the next nodes (blocks), Simulation for evaluating the quality of generated nodes, and Revision for updating the nodes with low scores.

### 3.2.1. SELECTION: GREEDY SELECTION.

DiffuReason recursively selects child nodes from the root until reaching an unexpanded node. Based on the value function V(s) from Section 3.2.3, we adopt a greedy strategy to select the node with the top-k current value as the basis for expansion:

$$s_t^* = \arg\max_{s_t \in \mathcal{C}_t} V(s_t) \tag{5}$$

where $C_t$ is the candidate node set. We posit that $s^*$ contains the currently optimal problem-solving idea (Reasoning Path), providing a reliable context for subsequent procedures.

### 3.2.2. EXPANSION: BLOCK-WISE DIFFUSION WITH DYNAMIC TRUNCATION

For the selected node $s^*$, the generator $G_\theta$ generates next timestep blocks following the generation process described in Section 3.1.1. To boost the diversity of exploration, $G_\theta$ samples $K$ times to obtain $K$ results, and each generated result becomes a child node in the tree. So the candidate set of timestep $t$ is the previous state $s_{t+1}$ plus newly generated blocks, which is,

$$C_t = s_{t+1} \cup \{b_t^1, b_t^2, \cdots, b_t^K\} \tag{6}$$

To fit the fixed-length text characteristics of the diffusion model, we previously set an indefinite length for each block in Section 3.1.3. However, this would cause the complete sentences output by the model to be truncated within fixed-length blocks. Therefore, we introduce a dynamic block-length adjustment mechanism: if the sentence length is shorter than the fixed length, the unused tokens are merged into the next block; if it is longer than the fixed length, it borrows length from the next block. This mechanism ensures

that every new expansion node in the search tree contains a semantically complete reasoning step.

### 3.2.3. SIMULATION: MULTI-DIMENSIONAL VALUATION VIA LOOKAHEAD

After generating candidate nodes, standard MCTS requires value evaluation. To avoid myopic behavior from a local perspective, we propose a multi-dimensional value function combining immediate reward and lookahead simulation.

For any state $s$, its value $V(s)$ is defined as:

$$V(s) = (1 - \lambda) \cdot R_p(s) + \lambda \cdot \mathbb{E}[R_o(\tau|s)] \qquad (7)$$

**Immediate Process Score ($R_p$).** Provided by the Process Reward Model $M_{PRM}$, evaluating the logical rigor of the current thought block (in-process node in the tree):

$$R_p(s_t) = M_{PRM}(b_t|s_{t-1}) \qquad (8)$$

**Lookahead Simulation Value ($R_o$).** To foresee whether the current path leads to a correct answer, we execute Monte Carlo Rollout. Starting from state $s$, we use a Fast Generation strategy to quickly estimate the outcome. Instead of full iterative denoising steps, we directly perform a 1-step (or few-step) greedy prediction over the remaining masked tokens to obtain the terminal prediction $x_{t-e}$ (where $t - e$ stands for time $t$ to the end):

$$R_o(\tau|s) = M_{ORM}(x_{t-e}) \qquad (9)$$

$$x_{t-e}^i = \begin{cases} x_t^i, & \text{if } x_t^i \neq [\text{M}] \\ \arg\max_{v \in \mathcal{V}} \log p_\theta(v|x_t), & \text{if } x_t^i = [\text{M}] \end{cases} \qquad (10)$$

We execute $M$ simulations (by introducing Gumbel noise to sample $M$ distinct terminal sequences $x_{t-e}$) and take the average score from Outcome Reward Model $M_{ORM}$ as a less biased estimate of future value.

### 3.2.4. REVISION: IN-PLACE REVISION VIA RE-MASKING.

Although $s_t^*$ is the current optimal choice, in complex mathematical reasoning, correct ideas may still be accompanied by local calculation errors. So we introduce the Confidence-aware Revision mechanism after the simulation.

**Low-confidence Localization.** We inspect the confidence of each token within $s_t^*$. First, we define the confidence score $\text{Score}(x_i)$ of a token $x_i$ as its generation probability, which is computed via a standard Softmax over the vocabulary logits from the diffusion generator. Rather than sampling from a probability distribution for re-masking, we deterministically select the tokens with the lowest confidence scores via a top-$k$ operation. Specifically, we identify

the index set $\mathcal{I}_{re}$ of tokens to be re-masked as follows:

$$\mathcal{I}_{re} = \arg\min_{\mathcal{I} \subset \{1,...,M\}, |\mathcal{I}|=k} \sum_{i \in \mathcal{I}} \text{Score}(x_i) \qquad (11)$$

where $\text{Score}(x_i) = \text{Softmax}(\text{logits})_{x_i}$, $M$ is the number of newly generated tokens in the block, and $k$ is the number of tokens to re-mask (determined by a predefined re-masking ratio).

**Diffusion-based Correction.** Based on the selected set $\mathcal{I}_{re}$, we construct a new mask vector $M_t'$ over the whole length. For any token index $i$ within the newly generated blocks, if $i \in \mathcal{I}_{re}$, the corresponding mask value in $M_t'$ is set to 0 (to be regenerated); otherwise, it is set to 1 (to be retained). Here, $M_t'$ can be regarded as an additional timestep between $t$ and $t - 1$, and we perform a new sampling process to obtain a new node:

$$x_t' = [\mu_\theta(x_t, t) + \sigma_t z] \odot M_t' \qquad (12)$$

This process utilizes the bidirectional attention mechanism of diffusion models to targetedly repair specific calculation errors or logical loopholes, resulting in a final high-quality node $s_t'$ instead of $s_t$ in the tree.

### 3.2.5. COMPARISON WITH STANDARD MCTS.

**UCB Rule.** DiffuReason no longer uses the UCB (Upper Confidence Bound) rule that is commonly employed in MCTS, meaning it no longer balances node exploration and exploitation. Instead, it directly selects the top-$k$ nodes with the highest scores. This saves memory for recording UCB values and eliminates the extra time overhead caused by Backpropagation (the fourth step in MCTS). Although DiffuReason loses part of the stochastic exploration mechanism, the revision mechanism allows low-confidence regions to be regenerated randomly, which compensates for the lack of exploration. DiffuReason will also trigger regeneration when its value is lower than a threshold, suppressing the expansion of low-value nodes.

**Fast Generation.** For efficiency considerations, DiffuReason employs a fast generation strategy during the simulation phase. Specifically, it sets the diffusion timestep directly to 1 for generation, rather than using multiple time steps for gradual denoising as in standard diffusion. Although this introduces some generation error, the effect is acceptable because the simulation operates on intermediate steps, and already generated regions effectively guide the generation of the remaining areas. The goal of the simulation is to evaluate the quality of the currently generated block; even if the quality of subsequent blocks is not particularly high, averaging over multiple samples can still reflect the quality of the current block well. The analysis in the experimental section shows that the fast generation strategy performs comparably

to the fine-grained multi-step generation strategy, while also improving generation efficiency.

**Revision.** Auto-Regressive (AR) models face a "rewriting cost" when correcting errors: due to causal attention constraints, if an error occurs at step $i$, the model must discard and regenerate step $i$ and all subsequent content $(i, i+1, \ldots, N)$. Assuming the cost of generating one token is $C_{gen}$, the cost of correcting a deep error is linear $O((N-i) \cdot C_{gen})$. In contrast, DiffuReason leverages the bidirectional attention mechanism of diffusion models to achieve $O(1)$ complexity for In-place Revision. When we detect an error in the $i$-th block, we only need to re-mask and regenerate the low-confidence tokens within that block, without re-computing subsequent blocks (assuming context semantic compatibility). Assuming the re-masking ratio for revision is $\gamma \in (0, 1)$ and block length is $M$, the revision cost for DiffuReason is:

$$Cost_{Refine} \approx \gamma \cdot M \cdot C_{step} \ll Cost_{Regenerate} \quad (13)$$

where $CstepC_{step}$ is the cost of diffusion denoising steps. This indicates that DiffuReason can continuously improve the quality of the reasoning path at a very low marginal cost, demonstrating extremely high data efficiency.

# 4. Experiment

## 4.1. Experimental Setup

**Datasets.** We selecte four mathematical reasoning datasets with increasing difficulty levels to comprehensively evaluate our method: GSM8K (Cobbe et al., 2021), which consists of elementary school mathematical problems; MATH500 (Hendrycks et al., 2021), a subset of high school competition problems covering algebra and geometry; Gaokao 2023 (Zhong et al., 2024), a comprehensive collection of mathematical problems from the 2023 Chinese College Entrance Examination, American Mathematics Competitions (AMC), and U.S. college entrance exams (SAT/ACT); and AIME 2024 (AI-MO, 2024), which involves high-difficulty Olympiad-level reasoning.

**Baselines.** We choose 2 widely-used diffusion models as the base LLM, which are LLaDA-8B (Nie et al., 2025a) and WeDLM-8B (Liu et al., 2025). For auto-regressive LLMs, we compare our method against several strong baselines, including: DeepSeekMath-7B-RL (), Llama3.1-8B-Instruct (Meta, 2024), and its Chain-of-Thought (CoT) variants (Wei et al., 2022b). For diffusion-based models, we include D1-LLaDA (Zhao et al., 2025), a diffusion reasoning model fine-tuned via reinforcement learning, and R3, a chain-structured approach with corrective capabilities, as two strong competitive baselines

**Implementation Details.** In Expansion, each node generates 3 candidates, and the candidate block length is 64. In simulation, both the Proposal Reward Model (PRM) and Outcome Reward Model (ORM) use the Qwen2.5-Math-PRM-7B model (Yang et al., 2024), and the weighting factor for the PRM score is set to $\lambda = 0.4$. In fast generation, we reduce 128 steps to 32 steps in diffusion denoising steps. The condition for triggering regeneration is 0.5. In Revision, the tokens with the 20% lowest probability are selected to be masked for regeneration.

**Metrics.** We report *Accuracy (%)* as the primary metric. We also record the average inference time per problem (*Avg. Time (s)*) to evaluate the efficiency of baselines.

## 4.2. Overall Performance

Table 1 presents the main experimental results of DiffuReason across different base models. Here, the results of R3 are inconsistent with those reported in the paper because the original paper reported 127 samples, whereas we ran the full test set. Additionally, there were bug issues in the source code downloaded from the paper; after we fixed them, the results we obtained might have introduced errors.

**Accuracy.** Regarding reasoning capability, we use three base models. It can be observed that since they are not specifically designed for mathematical data reasoning tasks, neither autoregressive nor non-autoregressive models hold an absolute advantage. Further training via reinforcement learning on both autoregressive and non-autoregressive models can significantly enhance reasoning ability. However, these improvements are not free: both DeepSeekMath-7B-RL and D1-LLaDA require additional training time and resources, and D1-LLaDA additionally necessitates a complete chain-of-thought as supervision signals, which are often unavailable in real-world reasoning tasks. Most training-free methods enhance reasoning capability through search, which indeed achieves better results compared to the base models. Notably, LLAMA3.1-8B-INSTRUCT with chain-of-Thought (CoT) shows significant improvement across multiple datasets. Nevertheless, search-based methods still cannot surpass the performance of retraining-based approaches.

We tested our search algorithm on two diffusion-based models. The results show that compared to the base models themselves, DIFFUREASON (LLaDA) achieved a minimum improvement of 1.8% and a maximum improvement of 3.8% across the four datasets; for DIFFUREASON (WEDLM), except for the relatively smaller gain on GSM8K, improvements exceeded 7% on all other datasets. Notably, on the more challenging AIME2024 dataset, our method achieved a 2-fold improvement. Compared to other search algorithms, our approach slightly underperformed the Chain-of-Thought (CoT) method with autoregressive models on two relatively simple datasets, but outperformed all autoregressive models on the more difficult datasets. Overall, our method achieves

*Table 1.* Main Results of **Accuracy** and **Efficiency** on Mathematical Reasoning Benchmarks. We apply DiffuReason to both LLaDA-8B and WeDLM-8B base models. **Bold** indicates the best results, and underline indicates the second-best results.

| MODEL | ARCH. | GSM8K | MATH500 | AIME2024 | GAOKAO2023 | TIME (S) |
|---|---|---|---|---|---|---|
| LLAMA3.1-8B-INSTRUCT | AR | 65.4 | 39.6 | 13.3 | 23.9 | 51.5 |
| LLADA-8B | DIFFUSION | 71.8 | 31.4 | 6.7 | 24.0 | 33.4 |
| WEDLM-8B | DIFFUSION | 89.2 | 58.8 | 3.3 | 43.6 | 18.3 |
| DEEPSEEKMATH-7B-RL | AR+RL | 88.2 | 52.4 | 3.3 | 43.6 | 23.6 |
| D1-LLADA | DIFF+RL | 82.1 | 40.2 | – | – | – |
| LLAMA3.1-8B-INSTRUCT + CoT | AR+SEARCH | 79.8 | 45.6 | 3.3 | 24.9 | 112.7 |
| R3-LLADA | DIFF+SEARCH | 72.3 | 31.9 | 6.7 | 15.1 | 41.8 |
| **DIFFUREASON (LLADA)** | DIFF+SEARCH | 75.6 | 34.6 | 10.0 | 25.8 | 88.3 |
| **DIFFUREASON (WEDLM)** | DIFF+SEARCH | **90.1** | **68.2** | **20.0** | **51.2** | 70.6 |

state-of-the-art performance, effectively validating its capability to enhance the reasoning abilities of baseline diffusion models.

**Efficiency.** When comparing reasoning capability and efficiency simultaneously, we observe that while our method incurs longer inference time compared to the base models, it demonstrates stronger reasoning capability and requires less time (save 37.3% comparing with LLAMA3.1-8B-INSTRUCT +CoT) than search algorithms based on auto-regressive models. This indicates that our algorithm effectively leverages the high efficiency inherent to diffusion models while significantly enhancing their reasoning abilities.

### 4.3. Ablation Studies

*Table 2.* Ablation Study on MATH500 and Gaokao2023.

| METHOD | MATH500 | GAOKAO2023 |
|---|---|---|
| DIFFUREASON (FULL) | **68.2** | **51.2** |
| W/O SIMULATION | 64.6 | 46.4 |
| W/O REVISION | 63.0 | 48.1 |

To verify the contribution of each component in DiffuReason, we conducted ablation studies using WeDLM as the base model on the MATH500 and Gaokao2023 datasets. The results are shown in Table 2. The results highlight the importance of lookahead valuation. The performance drop indicates that relying solely on the current step's PRM score is often myopic. The rollout mechanism endows the model with the critical ability to foresee dead ends. Furthermore, the backtracking mechanism serves as a safety net. Removing Revision resulted in the most severe performance degradation (-5.2%), as the model could not recover after straying into incorrect paths, leading to error accumulation.

### 4.4. Analysis and Discussion

#### 4.4.1. EFFECTIVENESS OF FAST GENERATION

To investigate the relationship between generation steps and performance, we compared the performance of Fast Generation versus Fine Generation on the MATH500 subset (20 problems from each difficulty level) using LLaDA-8B as the base model. We adjusted the computational budget by controlling the number of rollout iterations.

*Table 3.* Accuracy and efficiency across difficulty levels.

| METRIC | STEPS = 32 | STEPS = 128 |
|---|---|---|
| LEVEL 1 ACC (%) | 80.0 | 80.0 |
| LEVEL 2 ACC (%) | 55.0 | 50.0 |
| LEVEL 3 ACC (%) | 30.0 | 35.0 |
| LEVEL 4 ACC (%) | 15.0 | 20.0 |
| LEVEL 5 ACC (%) | 5.0 | 10.0 |
| OVERALL ACC (%) | 37.0 | **39.0** |
| AVG. TIME (S) | **86.9** | 136.7 |

As shown in Table 3, for simpler problems (Level 1–3), there is little difference in accuracy between fast generation and fine generation; however, for more difficult problems (Level 4–5), fast generation leads to an accuracy drop of about 5%. Nonetheless, compared to fine generation, fast generation achieves 1.6 times improvement in reasoning speed. We argue that this speed advantage is sufficient to compensate for the slight loss in accuracy. DiffuReason introduces a revision mechanism as compensation: when the accuracy of the fast generation output falls below a threshold, the system automatically triggers a revision process to refine the result through re-masked generation.

#### 4.4.2. BLOCKS-WISE SEARCHING

To demonstrate the effectiveness of our block-wise generation strategy, we design a control group: Random Node

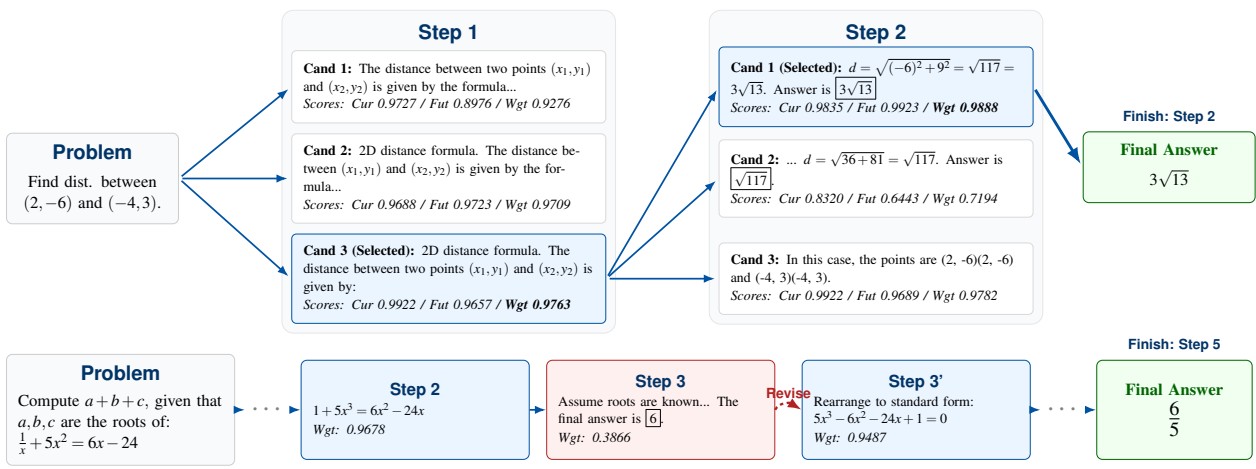

*Figure 2.* Illustration of a **Easy** and a **Hard** problem with revision. The model generates the correct derivation path, where high confidence scores (Wgt) are maintained throughout the process. To present the revision process in more detail, we omit multiple candidate answers for each step of the hard problem and only retain the one with the highest score.

*Table 4.* Block-wise strategy across Difficulty Levels

| METRIC | RANDOM | BLOCK-WISE |
|---|---|---|
| LEVEL 1 ACC (%) | 18.6 | 80.0 |
| LEVEL 2 ACC (%) | 17.8 | 55.0 |
| LEVEL 3 ACC (%) | 12.4 | 30.0 |
| LEVEL 4 ACC (%) | 6.3 | 15.0 |
| LEVEL 5 ACC (%) | 1.5 | 5.0 |
| OVERALL ACC (%) | 9.4 | 37.0 |
| AVG. TIME (S) | 8.7 | 86.9 |

Search. Instead of generating sequential blocks (Block-wise in Table 4), we randomly sample positions (Random in Table 4) to fill the entire answer as search nodes.

It shows that DiffuReason's block-wise generation is far superior to random generation. This is because in long-chain reasoning, the dependency between forward and backward processes is crucial. While random generation allows the model to output tokens with relatively high confidence, correcting errors in the generated result requires resampling, which consumes more resources. This is consistent with the observations in other works (Feng et al., 2025).

*Table 5.* Impact of Block Length on Performance.

| Length | Acc (%) | Avg. time (S) |
|---|---|---|
| 32 | 28.0 | 125.6 |
| **64 (Ours)** | 37.0 | 86.9 |
| 128 | 30.0 | 72.7 |

### 4.4.3. CASE STUDY

We present two cases in the Figure 2, each analyzing how the model infers on problems of different difficulty levels.

**Easy vs. Hard Problems.** We present a comparison between an easy problem and a hard problem in Figure 2, which is answered through a 2-layer and 5-layer search, respectively. This indicates that DiffuReason can flexibly adjust the search depth. Specifically, for linear logic problems, DiffuReason often locks onto the correct path in the first round of search, causing the Searcher to degrade into a simple verifier without triggering backtracking. Conversely, for complex theory problems, we observe that the search tree becomes deeper with more branches. The model utilizes Lookahead multiple times at intermediate steps to identify potential calculation errors and actively prunes them.

**The Effectiveness of Revision.** For complex problems, the candidate with the highest score may also not meet the threshold. This will trigger the revision mechanism in DiffuReason, which generates the current block through resampling and local token masking, bringing new exploration directions to the entire reasoning process. As shown at the bottom of Figure 2, the score of 3-th step is below the threshold. DiffuReason added step $3'$ and ultimately obtained the correct result in the 5-th step.

### 4.4.4. BLOCK LENGTH ANALYSIS

The length of the thought block is a key hyperparameter affecting search efficiency. We tested the impact of different block lengths (32, 64, 128 tokens) on the MATH500 dataset (20 problems from each difficulty level). As shown in Table 5, we choose 64 as the block length as it achieves the best

performance in general.

## 5. Conclusion

In this paper, to achieve a model with both strong reasoning capability akin to auto-regressive models and high reasoning efficiency like diffusion models, we propose DiffuReason, which is based on diffusion models. DiffuReason incorporates the core skills of Monte Carlo search used in auto-regressive models: search, evaluation, and revision. To enable searchability in the reverse denoising generation process of diffusion models, DiffuReason models sequence generation over time as a Markov Decision Process (MDP). For evaluating each reasoning step, DiffuReason introduces an evaluation method that combines the current state with future states. To facilitate content revision during generation, DiffuReason proposes a trigger mechanism for re-masking and sampling. Experimental results demonstrate that DiffuReason outperforms baseline diffusion models in reasoning capability and achieves a better balance between accuracy and efficiency compared to auto-regressive models. This demonstrates that our method can effectively integrate the accuracy of slow thinking from autoregressive models with the efficient fast generation of diffusion models.

## Acknowledgements

This work was partially supported by National Natural Science Foundation of China (NSFC Grant No. 62576353), the NSF for Distinguished Young Scholars under Grant No. 62425607 and 62125604, and the Key NSF of China under Grant No. 62136005.

## Impact Statement

This paper presents work whose goal is toadvance the field of MachineLearning. There are many potentialsocietal consequences of our work, nonewhich we feel must be specifically highlighted here.

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

# A. Extended Evaluation and Ablation Studies

In this section, we provide extended evaluation results and ablation studies conducted during the rebuttal phase to further validate the generalization, complementarity, and robustness of the DiffuReason framework.

## A.1. Complementarity with RL Post-Training

To investigate whether our inference-time search algorithm can act complementary to Reinforcement Learning (RL) post-training, we applied DiffuReason on top of d1-LLaDA, a model fine-tuned via RL. As shown in Table 6, our algorithm yields significant additional performance boosts across all difficulty levels of the MATH500 subset. The overall accuracy improved from 27.3% to 38.4%. This demonstrates that explicit test-time planning (System 2) is highly complementary to the implicit reasoning patterns acquired through RL fine-tuning (System 1).

*Table 6.* Performance on MATH500 Subset using an RL-finetuned base model (d1-LLaDA).

| METRIC | D1-LLaDA | + DIFFUREASON (OURS) |
|---|---|---|
| LEVEL 1 ACC (%) | 57.9 | 89.5 |
| LEVEL 2 ACC (%) | 42.9 | 52.4 |
| LEVEL 3 ACC (%) | 15.8 | 36.9 |
| LEVEL 4 ACC (%) | 14.3 | 9.5 |
| LEVEL 5 ACC (%) | 5.3 | 5.3 |
| OVERALL ACC (%) | 27.3 | **38.4** |
| AVG. TIME (S) | 30.1 | 89.2 |

## A.2. Ablation on Dynamic Block-Length Adjustment

To ensure the semantic atomicity of the thought blocks during expansion, we introduced a dynamic block-length adjustment mechanism that truncates blocks at logical delimiters (e.g., '\n' or '.'). We compared this against a baseline that strictly cuts off at a fixed token length (64 tokens). As shown in Table 7, our dynamic adjustment significantly improves accuracy (from 29.3% to 37.4%) and speeds up inference by reducing reasoning fragmentation.

*Table 7.* Ablation on Dynamic Block-Length Adjustment.

| METRIC | FIXED TOKEN LENGTH | DYNAMIC BLOCK (OURS) |
|---|---|---|
| LEVEL 1 ACC (%) | 78.9 | 84.2 |
| LEVEL 2 ACC (%) | 33.3 | 52.4 |
| LEVEL 3 ACC (%) | 31.6 | 31.6 |
| LEVEL 4 ACC (%) | 4.8 | 14.3 |
| LEVEL 5 ACC (%) | 0.0 | 5.3 |
| OVERALL ACC (%) | 29.3 | **37.4** |
| AVG. TIME (S) | 116.4 | 84.9 |

## A.3. Ablation on Value Function Components (w/o PRM)

To isolate the contribution of the search structure itself from the strength of the external Process Reward Model (PRM), we conducted an ablation study by removing the PRM from our value function, relying solely on the sparse Outcome Reward Model (ORM) via fast rollouts. As shown in Table 8, even without the fine-grained guidance of a PRM, DiffuReason achieves 34.3% accuracy, providing an 8.0% absolute improvement over the base model (26.3%). This proves that our tree expansion and simulation mechanisms inherently possess strong exploration capabilities.

*Table 8.* Ablation on removing the Process Reward Model (PRM).

| METRIC | W/O PRM (ORM ONLY) | OURS (PRM+ORM) |
|---|---|---|
| LEVEL 1 ACC (%) | 78.9 | 84.2 |
| LEVEL 2 ACC (%) | 42.9 | 52.4 |
| LEVEL 3 ACC (%) | 36.8 | 31.6 |
| LEVEL 4 ACC (%) | 14.3 | 14.3 |
| LEVEL 5 ACC (%) | 0.0 | 5.3 |
| OVERALL ACC (%) | 34.3 | **37.4** |
| AVG. TIME (S) | 78.6 | 84.9 |

## A.4. Performance Breakdown by Difficulty Levels

We present a detailed breakdown of the relative accuracy boost provided by DiffuReason across different problem difficulty levels in the MATH500 subset. As shown in Table 9, our Lookahead Search algorithm brings significant improvements across all levels. Notably, for the hardest Level 5 problems, the base model completely fails (0.0%), whereas our method successfully solves a portion of them (5.3%) by actively foreseeing dead ends and correcting local errors via in-place revision.

*Table 9.* Relative boost in accuracy across problem difficulty levels.

| METRIC | BASE MODEL | OURS | RELATIVE BOOST |
|---|---|---|---|
| LEVEL 1 ACC (%) | 52.6 | 84.2 | + 60.1% |
| LEVEL 2 ACC (%) | 42.9 | 52.4 | + 22.1% |
| LEVEL 3 ACC (%) | 26.3 | 31.6 | + 20.2% |
| LEVEL 4 ACC (%) | 9.5 | 14.3 | + 50.5% |
| LEVEL 5 ACC (%) | 0.0 | 5.3 | - |
| OVERALL ACC (%) | 26.3 | **37.4** | **+ 42.2%** |
| AVG. TIME (S) | 31.9 | 84.9 | - |

