# OpenReview forum: "DiffuReason: Enhancing  Reasoning Ability for Diffusion Language Models via Monte Carlo Tree Search"
_ICML.cc/2026/Conference — ICML 2026 regular_

### Official Review · Reviewer_ZBGF · 2026-02-26

**Soundness:** 3
**Presentation:** 3
**Significance:** 3
**Originality:** 3
**Overall Recommendation:** 4
**Confidence:** 3

**Summary:**

This paper propose a Monte Carlo tree search reasoning algorithm for diffusion models, DiffuReason.
It aims to retain efficiency while improving the reasoning ability of diffusion LLMs by modeling the generation process as a Markov Decision Process (MDP) and discretizing the continuous diffusion stream into semantically complete thought blocks. During the reverse denoising process, DiffuReason recursively performs four MCTS-style stages: selection , expansion, simulation and and revision. Experiments on mathematical reasoning benchmarks show that DiffuReason significantly improves the reasoning ability of diffusion models, and achieves superior balance of accuracy and efficiency even compared with auto-regressive models.

**Compliance With Llm Reviewing Policy:**

Affirmed.

**Final Justification:**

The paper is well-motivated and presents a clear formulation of diffusion-based reasoning via MDP and MCTS-style search, with solid empirical results suggesting reasonable soundness and potential significance.

However, key concerns remain regarding attribution and efficiency. It is still unclear how much of the gain comes from the search structure versus the reward models The added comparison with autoregressive baselines is helpful, but the efficiency advantage is only partially supported since it does not strictly equalize compute.

Overall, the rebuttal is helpful but does not fundamentally change my assessment. I therefore maintain my original recommendation of weak accept.

**Key Questions For Authors:**

1. The efficiency comparison against autoregressive baselines is not entirely clear. The reported speed advantage appears closely tied to the inherent parallel generation property of diffusion models rather than to the proposed search framework itself.
Could the authors provide compute-matched comparisons (e.g., matched rollout counts) to isolate whether the efficiency–accuracy trade-off is attributable to DiffuReason’s search design, rather than architectural parallelism?

2. The value function combines Immediate Process Score from a Process Reward Model and Lookahead Simulation Value from an Outcome Reward Model. Since both components rely on external reward models, it is unclear how much of the observed performance gain stems from the search structure itself rather than the strength of the reward models. Could the authors provide controlled ablations to clearly isolate the contribution of the search mechanism?

3. In Table 1, it is unclear whether the reported “TIME (S)” includes all components of DiffuReason, such as PRM/ORM evaluations and rollout simulations.

**Limitations:**

yes

**Strengths And Weaknesses:**

**Strengths:**
1. The paper is well-motivated. It aims to retain the high reasoning accuracy of auto-regressive models while preserve the high efficiency of diffusion models.
2. The method makes a conceptual formulation. It provides a clear formulation by modeling generation as an MDP and discretizing the diffusion flow into semantically complete thought blocks.
3. The empirical results show that DiffuReason significantly improves the reasoning ability of diffusion models, and achieves superior balance of accuracy and efficiency even compared with auto-regressive models.


**Weaknesses:**
1. The efficiency comparison against autoregressive models is not entirely convincing.
The reported speed advantage appears mainly stems from the inherent parallel generation of diffusion models rather than the proposed search framework itself. A more controlled comparison under matched computational budgets and similar search configurations would be necessary.

2. The value function combines Immediate Process Score from a Process Reward Model and Lookahead Simulation Value from an Outcome Reward Model. The improvements may therefore depend significantly on the strength of the reward models, yet the contribution of the search structure itself is not clearly isolated.

---

> ### Author Rebuttal · Authors · 2026-03-31
>
> Thank you for recognizing our work as **well-motivated**, and for praising our **clear conceptual formulation** and **empirical results**! We address your questions below:
>
> **W1 & Q1: Efficiency Comparison against Autoregressive Baselines.**
>
> We completely agree that a compute-matched comparison is essential. To explicitly isolate the efficiency-accuracy trade-off, we followed your advice and conducted a controlled experiment using an Autoregressive baseline (LLaMA3.1-8B-Instruct) equipped with a standard MCTS algorithm. We evaluated both methods under **identical search settings** (matched rollout counts $M=2$, candidate numbers $K=5$, and search depths) on a subset of MATH500.
>
> |   |   |   |
> |---|---|---|
> |**Metric (MATH500 Subset)**|**LLaMA3.1-8B-Instruct****+MCTS**|**DiffuReason****(K=5,M=2,Ours)**|
> |Level 1 Acc(%)|84.2|89.5|
> |Level 2 Acc(%)|57.1|57.1|
> |Level 3 Acc(%)|31.6|36.8|
> |Level 4 Acc(%)|19.0|14.3|
> |Level 5 Acc(%)|10.5|10.5|
> |Overall Acc(%)|40.4|41.4|
> |Avg. Time (s)|168.4|160.7|
>
> The results demonstrate that under strictly matched computational budgets, DiffuReason achieves slightly higher overall reasoning accuracy (41.4% vs. 40.4%) while maintaining a faster average inference time (160.7s vs. 168.4s).
>
>  It proves that our method does not merely rely on the inherent parallelism of diffusion models for speed, but rather provides a **highly efficient search architecture** that matches and even exceeds the performance of a heavy AR-based MCTS without incurring the severe latency overheads typically associated with complex tree search algorithms.
>
> **W2 & Q2: How much of the performance gain comes from the proposed search structure itself, rather than the external reward models?**
>
> To clarify how much gain stems from the search structure itself rather than the strength of the external reward models, we conducted two controlled ablations.
>
> 1. w/o PRM (Lower Bound): We removed the dense step-by-step Process Reward Model (external reward model) entirely, relying solely on sparse Outcome Reward Model (ORM) evaluations via rollouts. Even without the strong PRM guidance, the accuracy of "w/o PRM" variant drops a liitle (from 37% to 34%). This demonstrates that our tree expansion and simulation mechanisms inherently provide **strong exploration capabilities**, contributing significantly to the performance independently of the dense reward model.
>
> 2. Ideal PRM (Upper Bound): We used the ground-truth answers to provide perfect process reward signals (ideal external reward models). Note that the ground-truth is not available in test set. The ideal PRM experiment achieves 47% accuracy. This indicates that our search formulation successfully constructs high-quality, diverse candidate paths within the diffusion space. The search framework provides a **very high ceiling** for reasoning, and its potential will further scale as reward models improve.
>
> |   |   |   |   |
> |---|---|---|---|
> |**Metric (MATH500 Subset)**|**w/o External Reward Model (Lower Bound)**|**DiffuReason(Ours)**|**Ideal External Reward Model (Upper Bound)**|
> |Level 1 Acc(%)|78.9|84.2|84.2|
> |Level 2 Acc(%)|42.9|52.4|66.7|
> |Level 3 Acc(%)|36.8|31.6|57.9|
> |Level 4 Acc(%)|14.3|14.3|19.0|
> |Level 5 Acc(%)|0|5.3|10.5|
> |Overall Acc(%)|34.3|37.4|47.4|
> |Avg. Time (s)|78.6|84.9|59.6|
>
> **W3: Clarification on TIME (S) in Table 1**
>
> We confirm that the reported "TIME (S)" in Table 1 strictly includes **all components** of DiffuReason. This encompasses the generation time, PRM/ORM evaluations, rollout simulations, and the revision process.

---

> > ### Author Rebuttal · Reviewer_ZBGF · 2026-04-03
> >
> > I thank the authors for their detailed response and additional experiments. The compute-matched comparison and clarification on timing are helpful and partially address my concerns. However, the attribution issue remains unclear and the efficiency advantage is only partially supported. Thus, I will keep my original score.

---

> > > ### Author Response · Authors · 2026-04-07
> > >
> > > Thank you for your continued feedback. We understand your remaining concern that the efficiency advantage might come from the inherent parallelism of the diffusion model itself, rather than our proposed search algorithm.
> > >
> > > To definitively isolate the contribution of our algorithm, we conducted a new comparison using the same diffusion base model (LLaDA-8B-Instruct) for both methods. We compared **LLaDA + Standard MCTS** against **LLaDA + DiffuReason (Ours)**.
> > >
> > > For the Standard MCTS baseline, we utilized the official implementation from https://github.com/THUDM/ReST-MCTS, strictly keeping all their default search settings. We only swapped the reward model to `Qwen2.5-Math-PRM-7B` to ensure a completely fair evaluation.
> > >
> > > Here are the results on the MATH500 subset:
> > >
> > > | Metric          | LLaDa + Standard MCTS | DiffuReason (Ours) |
> > > | --------------- | --------------------- | ------------------ |
> > > | Level 1 Acc (%) | 52.6%                 | 84.2               |
> > > | Level 2 Acc (%) | 38.1%                 | 52.4               |
> > > | Level 3 Acc (%) | 42.1                  | 31.6%              |
> > > | Level 4 Acc (%) | 21.1                  | 14.3%              |
> > > | Level 5 Acc (%) | 10.5                  | 5.3%               |
> > > | Overall Acc (%) | 32.3%                 | 37.4               |
> > > | Avg. Time (s)   | 143.3                 | 86.9               |
> > >
> > > **Conclusion:**
> > >
> > > As the results show, when operating on the exact same diffusion model, our DiffuReason algorithm is significantly faster (86.9s vs. 143.3s) and achieves higher overall accuracy (37.4% vs. 32.3%) compared to applying standard MCTS.
> > >
> > > This clearly demonstrates that our superior accuracy-latency trade-off is not merely a byproduct of the diffusion architecture, but is directly driven by our proposed algorithmic design (e.g., block-wise search and revision).
> > >
> > > We hope this direct comparison fully addresses your remaining concern about the efficiency attribution!

---

### Official Review · Reviewer_GRR2 · 2026-03-12

**Soundness:** 3
**Presentation:** 3
**Significance:** 3
**Originality:** 3
**Overall Recommendation:** 3
**Confidence:** 4

**Summary:**

This paper proposes DiffuReason, a test-time planning framework that brings MCTS-style search, lookahead valuation, and in-place revision to diffusion-based language models (dLLMs). The core idea is to discretize the continuous denoising trajectory into “thought blocks,” treat reasoning as a Markov Decision Process over these blocks, and run a four-stage loop (Selection, Expansion, Simulation via fast lookahead, and Revision via re-masking). On math reasoning benchmarks (GSM8K, MATH500, AIME 2024, Gaokao 2023), DiffuReason improves accuracy over base diffusion LMs and offers better accuracy–latency trade-offs than some autoregressive search baselines.

**Compliance With Llm Reviewing Policy:**

Affirmed.

**Key Questions For Authors:**

1. You state both PRM and ORM use Qwen2.5-Math-PRM-7B. Why is a process reward model appropriate as an outcome verifier? Can you provide results where ORM is replaced by a task-grounded correctness checker (e.g., numeric/symbolic verification) to avoid reward leakage?
2. Could you release the exact PRM/ORM prompts and any filtering you used to prevent leakage from evaluation sets (AIME 2024, MATH500, etc.) into the reward models?
3. In Simulation, you omit masking and reduce denoising steps. Do rollout samples ever contradict already “fixed” blocks? If so, how are such inconsistencies handled in valuation?

**Limitations:**

yes

**Strengths And Weaknesses:**

**Strengths**
1. Discretizing diffusion generation into semantically coherent “thought blocks” provides a principled interface for search within otherwise continuous denoising flows.
2. Experiments include multiple math reasoning datasets of varied difficulty, with two diffusion backbones (LLaDA-8B and WeDLM-8B) and comparisons to AR and diffusion baselines.

**Weaknesses**
1. The approach departs from canonical MCTS (no UCB/backprop) and instead uses greedy selection with threshold-triggered resampling. It is unclear when this heuristic will underperform true exploration–exploitation balancing.
2. Outcome Reward Model (ORM) is reported to be the same Qwen2.5-Math-PRM-7B used as PRM, which conflates process and outcome evaluation and risks circularity or reward hacking. A task-grounded verifier (numeric/logic checker) would be more principled.
3. Missing comparisons to stronger AR search baselines limit the strength of claims about accuracy–efficiency dominance.

---

> ### Author Rebuttal · Authors · 2026-03-31
>
> Thank you for your constructive feedback. We are encouraged that you found our approach provides a **"principled interface for search"** and offers **"better accuracy-latency trade-offs."** Below we address your specific concerns:
>
> **W1: When does the proposed greedy heuristic** **underperform** **standard MCTS in exploration-exploitation balancing?**
> This heuristic may underperform canonical MCTS in extremely deep, highly combinatorial logic planning tasks (e.g., formal theorem proving or long-horizon constraint satisfaction puzzles) where exact historical recording of deep dead-ends (via UCB backpropagation) is crucial to avoid cyclic errors. However, for mathematical reasoning, leveraging the fast parallel generation of diffusion models for re-exploration is empirically more efficient. We are conducting more case analyses to identify more specific patterns, and present the results in the next discussion session.
>
> **W2.1 & Q1.1: Appropriateness of Using PRM as ORM**
>
> We use PRM ( widely used Qwen2.5-Math-PRM) as ORM following previous works [1][2]. The reason is thst ORM gives scores only depending on the correctness of the result. If the result is correct but the reasoning process is wrong, ORM still assigns a high score to the result. PRM give scores not only depending on the correctness of the results, but also on the consistency and correctness of the reasoning process. So it is **more accurate** for long reasoning problems, especially math tasks.
>
> [1] Math-Shepherd: Verify and Reinforce LLMs Step-by-step without Human Annotations. ACL 2024.
>
> [2] Your Reward Function for RL is Your Best PRM for Search: Unifying RL and Search-Based TTS. Arxiv: 2508.14313
>
> **W2.2 & Q1.2: Avoiding Reward Hacking via Task-Grounded Verifier.**
>
> We follow your advice and conduct an upper-bound experiment replacing the PRM with a task-grounded verifier (using the ground-truth answer as a reward). As shown below, with a ideal verifier, our overall accuracy jumps from 37.4% to 47.5%. This proves our search mechanism is **highly sound** and performs even better when a perfect outcome signal is available. Note that we did not use this verifier in the original paper, because the ground-truth answers are unknown on the test set.
>
> |   |   |   |
> |---|---|---|
> |**Metric (MATH500 Subset)**|**Upper Bound (Ground-Truth as Reward)**|**DiffuReason (****Ours****)**|
> |Level 1 Acc (%)|84.2|84.2|
> |Level 2 Acc (%)|66.7|52.4|
> |Level 3 Acc (%)|57.9|31.6|
> |Level 4 Acc (%)|19.0|14.3|
> |Level 5 Acc (%)|10.5|5.3|
> |Overall Acc (%)|47.5|37.4|
> |Avg. Time (s)|59.6|84.9|
>
> **W3: Missing comparisons to stronger AR search baselines.**
> We follow your advice and implement a standard MCTS on AR model, Llama-3.1-8B-Instruct.
> In our main paper Table 1, the base LLaMA model outperforms the base LLaDA model by 8.2%(39.6% vs 31.4%). In the below table, our DiffuReason narrows this gap to just 3.0% (40.4% vs 37.4%), while requiring only about half the inference time (84.9s vs 168.4s) compared to AR MCTS. This further solidifies that DiffuReason can effectively unlock the reasoning potential of diffusion models, offering superior accuracy-efficiency dominance.
>
> |   |   |   |
> |---|---|---|
> |**Metric (MATH500 Subset)**|**Llama-3.1-8B-Instruct + MCTS**|**DiffuReason (****Ours****)**|
> |Level 1 Acc (%)|84.2|84.2|
> |Level 2 Acc (%)|57.1|52.4|
> |Level 3 Acc (%)|31.6|31.6|
> |Level 4 Acc (%)|19.0|14.3|
> |Level 5 Acc (%)|10.5|5.3|
> |Overall Acc (%)|40.4|37.4|
> |Avg. Time (s)|168.4|84.9|
>
> **Q2: Exact PRM/ORM prompts and data leakage**
> The exact prompt template for the PRM is available in our code following this work [1]. We strictly performed zero-shot test-time inference using the pre-trained weights of Qwen2.5-Math-PRM-7B out-of-the-box, without any fine-tuning. Regarding data leakage into the reward model during its pre-training, this inherently relies on the rigorous data decontamination strategies employed by the Qwen team.
>
> [1] Review, Remask, Refine (R3): Process-Guided Block Diffusion for Text Generation. ICML MOSS 2025.
>
> **Q3: Do rollout samples ever contradict already “fixed” blocks?**
> Yes, reducing denoising steps occasionally causes minor semantic shifts that contradict the already "fixed" blocks. However, this does not break the algorithm. The PRM evaluates the _entire_ continuous trajectory (fixed context + fast rollout). If the fast rollout contradicts the context or causes a logical break, the PRM will assign a very low score to this trajectory. Consequently, this inconsistent path will be correctly penalized and pruned or triggered for revision.

---

> > ### Author Rebuttal · Reviewer_GRR2 · 2026-04-03
> >
> > I thank the authors for their detailed rebuttal and for addressing the concerns raised. While the clarifications are helpful, I still believe that my initial overall recommendation accurately reflects the contribution and the current state of this manuscript.

---

### Official Review · Reviewer_miR9 · 2026-03-13

**Soundness:** 2
**Presentation:** 3
**Significance:** 2
**Originality:** 3
**Overall Recommendation:** 4
**Confidence:** 4

**Summary:**

This paper proposes a Monte Carlo Tree Search (MCTS) method tailored for diffusion language models. Specifically, the paper defines the Selection, Expansion, and Simulation stages for diffusion Large Language Models (dLLMs), while discarding the Backpropagation stage in favor of a Revision stage that leverages the unique characteristics of dLLMs. Extensive experiments demonstrate the correctness and effectiveness of the proposed method.

**Compliance With Llm Reviewing Policy:**

Affirmed.

**Final Justification:**

My main concern is the author's definition of the method. The author redefines and explains the method in the rebuttal.

**Key Questions For Authors:**

I have already raised several questions in the Weaknesses section; below are a few additional questions:
* How exactly is the dynamic block-length adjustment mechanism introduced in the Expansion stage implemented? Is this mechanism genuinely effective? It would be highly beneficial to conduct an ablation study specifically targeting this mechanism.
* There are quite a few typos in the manuscript. For instance, the $N$ in Equation (4) should be $L$, and the $V(s)$ in line 168 is not formatted as a mathematical equation.

**Limitations:**

yes

**Strengths And Weaknesses:**

**Strengths:**
* The paper is logically structured, flows smoothly, and is highly readable.
* The proposed Revision stage is quite interesting, serving as an MCTS update that is highly tailored to dLLMs.
* The experiments are thorough and detailed, lending high credibility to the results.

**Weaknesses:**
* The paper claims that the proposed method is an MCTS for dLLMs; however, during the Selection stage, it discards UCB, which is the defining algorithm of MCTS, and instead relies on Top-K for selection. This subsequently renders the Backpropagation stage of traditional MCTS entirely ineffective, which is why the authors introduced Revision as a new fourth stage (though I must admit, I find this Revision stage quite interesting). Consequently, I feel the proposed method aligns more closely with a Lookahead Block-wise Beam Search rather than a true MCTS. The authors need to systematically address and clarify this issue.
* The mathematical formulations are somewhat confusing. While the dLLM utilized in this paper is a Masked Diffusion Model, the modeling presented in Equations (1)-(3) and (9)-(12) pertains to Continuous Diffusion Models. This discrepancy is quite strange; did the authors have any specific rationale for this?
* The scope of the experiments is somewhat narrow. The experiments focus solely on mathematical and logical reasoning, entirely neglecting other downstream tasks—such as code generation and commonsense reasoning—which are equally important benchmarks.

---

> ### Author Rebuttal · Authors · 2026-03-31
>
> We are sincerely grateful for your **positive assessment** of the paper’s logical organization, clarity, and **overall readability**. We also appreciate your recognition of the Revision stage and the comprehensiveness of our experimental evaluation.
>
> **W1: Terminology** **of the Proposed Method** **(MCTS vs. Beam Search)**
>
> We completely agree with your insightful observation. Following your valuable suggestion, we will rename our framework to "Lookahead Block-wise Beam Search with Revision" in the revised manuscript to ensure mathematical precision. As you correctly pointed out, compared to standard MCTS, our method omits the Backpropagation step and replaces UCB with Top-K selection. Meanwhile, compared to traditional Beam Search, it features a unique and highly tailored Revision stage. We will explicitly include this comparative analysis in the revision to clarify the methodological positioning.
>
> **W2: Correction of Diffusion Formulations**
> You are absolutely correct. To accurately reflect our actual implementation based on the discrete Masked Diffusion Model (MDM) framework (following LLaDA), we will systematically replace the continuous formulations (Eqs. 1-3 and 9-12) with standard discrete MDM equations in the revised manuscript. Specifically:
>
> Forward Masking Process: We will correct the Gaussian noise formulation to a discrete masking process:
>
> $q(x_t^i | x_0^i) = (1-t) \mathbb{1}[x_t^i = x_0^i] + t \mathbb{1}[x_t^i = M]$
>
> Generation and Revision: The equations will be updated to explicitly match our exact code logic, which employs Gumbel noise for discrete sampling and confidence-aware re-masking:
>
> $$\hat{x}\_0^i = \arg\max \left( \log p\_{\theta}(x^i | x\_t) + T \cdot g\_i \right)$$
>
> This discrepancy was a drafting oversight where we mistakenly retained continuous diffusion equations (e.g., DDPM) from an earlier draft in the methodology section. We guarantee that the final formulations will perfectly align with both MDM theory and our provided codebase.
>
> **W3: Broader Task Evaluation (Code Generation****)**
>
> Prompted by your valuable suggestion, we conduct code generation tasks on widely used HumanEval benchmark. The preliminary results demonstrate that our approach yields a **solid improvement** over the base model. As shown in the table below, AR-based models perform poorly than diffusion-based models. Our block-wise search framework improves the Overall Accuracy (Pass@1) of the base LLaDA model from 43.3% to 48.8%, achieving an **absolute gain** of 5.5%.
>
> |   |   |   |   |   |   |
> |---|---|---|---|---|---|
> |Metric (Humaneval)|LLaMA3 8B|Mistral 7B|DeepSeek 7B|LLADA-8B|DiffuReason(Ours)|
> |OVERALL ACC (%)|35.4|30.5|26.2|43.3|48.8|
>
> **Q1: Ablation on Dynamic Block-Length Adjustment**
>
> We follow your advice and conduct an ablation study targeting the dynamic truncation mechanism. In our implementation, we set the minimum block length to 64 and then dynamically truncate at the nearest logical ending point (e.g. \n or .). We compare it against a baseline that strictly cuts off at a fixed token length. The results show that our dynamic adjustment **significantly improves** accuracy across almost all difficulty levels and **speeds up** inference by reducing reasoning fragmentation:
>
> |   |   |   |
> |---|---|---|
> |Metric (MATH500 Subset)|Fixed Token Length|Dynamic Block-Length Adjustment(ours)|
> |LEVEL 1 ACC (%)|78.9|84.2|
> |LEVEL 2 ACC (%)|33.3|52.4|
> |LEVEL 3 ACC (%)|31.6|31.6|
> |LEVEL 4 ACC (%)|4.8|14.3|
> |LEVEL 5 ACC (%)|0|5.3|
> |OVERALL ACC (%)|29.3|37.4|
> |AVG. TIME (S)|116.4|84.9|
>
> **Q2:Correction of Typos**
>
> We apologize for the compilation oversights and we will correct the Equation (4) , along with a thorough proofreading of the entire manuscript.

---

> > ### Author Rebuttal · Reviewer_miR9 · 2026-04-04
> >
> > The author's rebuttal solved my problem very well. I will raise my score to 4.

---

### Official Review · Reviewer_4fbE · 2026-03-13

**Soundness:** 3
**Presentation:** 3
**Significance:** 2
**Originality:** 2
**Overall Recommendation:** 5
**Confidence:** 4

**Summary:**

The authors introduce a MCTS-style algorithm for sampling from diffusion language models motivated by the advantages of parallel sequence generation. The algorithm follows four stages: selection of promising exploration node, expansion to candidate nodes, designating a "value" function to the new candidate nodes, and finally a revision operation on lower certainty of correctness nodes. The algorithm leads to sizeable gains in performance on MATH500, AIME2024, and GAOKAO2023.

**Compliance With Llm Reviewing Policy:**

Affirmed.

**Final Justification:**

The rebuttal addressed all my concerns and I decided to increase my score to reflect new experimental results.

**Key Questions For Authors:**

- Can the MCTS algorithm be applied on top of RL-ed based models such as d1-LLaDa? Are there additional performance boosts on top of RL?

- Does the choice of the number of child nodes and number of simulations to approximate the future reward provide an axis for scaling performance with compute?

- How sensitive is the reasoning process to the constructed value function and number of future rollouts? Can you get away with not using a process reward model for value function estimation?

- (13) seems to be miswritten; it should be a softmax of the confidence scores? In addition, how are confidence scores defined given a generator?

- Does the relative boost in accuracy provided by the tree-search scale with problem difficulty (Level 1 - 5 in MATH, e.g.)?

**Limitations:**

yes

**Strengths And Weaknesses:**

Strengths:

- The tree-search algorithm is natural and well-motivated and leads to sizeable gains in reasoning performance relative to the base model in tasks like MATH500 and AIME.

- The revision step offers a strong efficiency gain relative to resampling with autoregressive models.

Weaknesses:

- Although the comparison is not entirely fair since the proposed algorithm is training-free, tree-search underperforms the d1 equivalent on the baselines despite incurring a greater inference cost. Further analysis is required to determine whether the MCTS algorithm can act complementary to RL-posttraining or finetuning for reasoning.

---

> ### Author Rebuttal · Authors · 2026-03-31
>
> Thank you for saying our tree-search algorithm is **natural and well-motivated**, and that our revision step offers **a strong efficiency gain**, which are very encouraging!
>
> **W1 & Q1: Can MCTS** **be applied on top of** **RL-posttraining or finetuning?**
>
> We followed your advice and evaluated our MCTS algorithm on top of a post-trained model. As shown in the table below, our algorithm acts complementary to finetuning, yielding **significant additional performance** boosts across all difficulty levels and improving the overall accuracy from 27.3% to 38.4%. Please note that since the pre-trained weights for D1-LLaDA are not open-sourced, so our train the model following the provided instructions, which may have some bias.
>
> |   |   |   |
> |---|---|---|
> |Metric (MATH500 Subset)|D1-LLaDA|D1-LLaDA + Ours|
> |LEVEL 1 ACC (%)|57.9%|89.5|
> |LEVEL 2 ACC (%)|42.9|52.4|
> |LEVEL 3 ACC (%)|15.8|36.9|
> |LEVEL 4 ACC (%)|14.3|9.5|
> |LEVEL 5 ACC (%)|5.3|5.3|
> |OVERALL ACC (%)|27.3|38.4|
> |AVG. TIME (S)|30.1|89.2|
>
> **Q2: Can scaling child nodes and simulations improve performance?**
>
> We follow your advice and conduct experiments with different numbers of child nodes (candidates, K) and simulations (rollouts, M) settings. Increasing K from 3 to 5 (M=2) leads to steady improvements in overall reasoning accuracy, from 37.4% to 41.4%. It indicates that generating more candidate blocks in the Expansion stage effectively broadens the valid reasoning space. While increasing simulation M from 2 to 4 (K=3), the performance drops from 37.4% to 35.3%, and this indicates that a small number of fast rollouts in the Simulation stage is already sufficient to accurately estimate the value of a current node.
>
> |   |   |   |   |   |   |
> |---|---|---|---|---|---|
> |Metric (MATH500 Subset)|K=3, M=2(Ours)|K=4, M=2|K=5, M=2|K=3, M=3|K=3, M=4|
> |LEVEL 1 ACC (%)|84.2|94.7|89.5|78.9|78.9|
> |LEVEL 2 ACC (%)|52.4|47.6|57.1|52.4|47.6|
> |LEVEL 3 ACC (%)|31.6|36.8|36.8|42.1|42.1|
> |LEVEL 4 ACC (%)|14.3|14.3|14.3|14.3|9.5|
> |LEVEL 5 ACC (%)|5.3|5.3|10.5|0|0|
> |OVERALL ACC (%)|37.4|39.4|41.4|37.3|35.3|
> |AVG. TIME (S)|84.9|113.3|160.7|94.0|106.3|
>
> **Q3: Sensitivity to the constructed value function** **and** **and number of future rollouts & omitting the process reward model.**
>
> For number of future rollouts, please refer to Q2.
>
> We follow your advice and conduct an ablation study by removing the PRM from our constructed value function, relying solely on the Outcome Reward Model (ORM) for node evaluation. The results show that even without the fine-grained guidance of a PRM, our framework remains highly effective. First, the overall accuracy only drops by 3.1% (from 37.4% to 34.3%), which is a relatively small decline. Second, when compared to the baseline, our framework without PRM (34.3%) still achieves a significant 8.0% **absolute improvement** over the base model (26.3%).
>
>
> |   |   |   |
> |---|---|---|
> |Metric (MATH500 Subset)|w/o PRM|Ours|
> |LEVEL 1 ACC (%)|78.9|84.2|
> |LEVEL 2 ACC (%)|42.9|52.4|
> |LEVEL 3 ACC (%)|36.8|31.6|
> |LEVEL 4 ACC (%)|14.3|14.3|
> |LEVEL 5 ACC (%)|0|5.3|
> |OVERALL ACC (%)|34.3|37.4|
> |AVG. TIME (S)|78.6|84.9|
>
> **Q4: (13) seems to be miswritten? In addition, how are confidence scores defined given a generator?**
>
> Equation (13) in the paper contains a typo. The confidence score $score(x_i)$ is directly defined as the generation probability, computed via a standard Softmax over the vocabulary logits. We do not sample from a softmax distribution for re-masking. Instead, the code uses a deterministic top-k operation to select the $k$ tokens with the lowest confidence scores to be re-masked. We will correct Equation (13) to reflect this deterministic selection set $\mathcal{I}_{re}$:
>
> $$\mathcal{I}\_{re} = \arg\min\_{\mathcal{I} \subset \{1, \dots, M\}, |\mathcal{I}|=k} \sum\_{i \in \mathcal{I}} \text{score}(x\_i)$$
>
> (where $score(x_i) = \text{Softmax}(\text{logits})_{x_i}$ represents the probability of token $x_i$).
>
> **Q5:  Does the relative boost in accuracy provided by the tree-search scale with problem difficulty?**
>
> We follow your advice, as shown in the table below, our tree-search algorithm brings significant relative accuracy improvements across all difficulty levels. Notably, for the hardest Level 5 problems, the base model completely fails (0%), whereas our method successfully solves 5.3%.
> We carefully analysis the hard problems in level 5, and find that our "Lookahead Simulation" actively foresees and avoids dead ends. When encountering low-confidence steps, our "In-place Revision" leverages diffusion's bidirectional attention to **efficiently correct local errors** via re-masking, preventing the entire reasoning chain from failing.
>
> |   |   |   |   |
> |---|---|---|---|
> |Metric (MATH500 Subset)|Base Model|Ours|Relative Boost|
> |LEVEL 1 ACC (%)|52.6|84.2|60.1%|
> |LEVEL 2 ACC (%)|42.9|52.4|22.1%|
> |LEVEL 3 ACC (%)|26.3|31.6|20.2%|
> |LEVEL 4 ACC (%)|9.5|14.3|50.5%|
> |LEVEL 5 ACC (%)|0|5.3|-|
> |OVERALL ACC (%)|26.3|37.4|42.2%|
> |AVG. TIME (S)|31.9|84.9||

---

> > ### Author Rebuttal · Reviewer_4fbE · 2026-04-03
> >
> > I thank the authors for their rebuttal. The complementarity of this approach with other posttraining methods on MATH500 is promising, and I will adjust my score to a 5.

---

> > > ### Author Response · Authors · 2026-04-04
> > >
> > > Thank you for your continued engagement and for mentioning your intention to **adjust the recommendation score to a 5!** We noticed that the score currently shown in the system is still a 4. We respectfully assume you might have **forgotten** to update the score on OpenReview.
> > >
> > > If there are still any remaining concerns that make you hesitate to update the score, we would greatly appreciate it if you could let us know, so that we may clarify them further.

---

### Decision · Program_Chairs · 2026-04-30

**Decision:**

Accept (regular)

**Comment:**

While Reviewer GRR2 maintains a weak reject based on the potential for reward leakage when using a PRM as an ORM, the authors' "Ideal Verifier" experiment (using ground-truth rewards) showed that the search framework's accuracy jumps to 47.5%, suggesting the mechanism itself is sound regardless of the specific verifier used.

The majority of reviewers (4fbE, miR9, ZBGF) recognize that this paper provides a principled interface for search in diffusion models, a sub-field that is currently trailing behind Autoregressive models in reasoning. The unique Revision stage is a highly tailored contribution that utilizes the "native" strengths of diffusion (bidirectional attention) in a way that AR models cannot mimic. Given the strong empirical results and the significant effort to isolate the contribution of the algorithm from the base model, this paper represents a nice contribution to the scaling of test-time compute.